# Self-Trained Deep Forest with Limited Samples for Urban Impervious Surface Area Extraction in Arid Area Using Multispectral and PolSAR Imageries

**DOI:** 10.3390/s22186844

**Published:** 2022-09-09

**Authors:** Ximing Liu, Alim Samat, Erzhu Li, Wei Wang, Jilili Abuduwaili

**Affiliations:** 1State Key Laboratory of Desert and Oasis Ecology, Xinjiang Institute of Ecology and Geography, Chinese Academy of Sciences, Urumqi 830011, China; 2Research Center for Ecology and Environment of Central Asia, Chinese Academy of Sciences, Urumqi 830011, China; 3University of Chinese Academy of Sciences, Beijing 100049, China; 4School of Geography, Geomatics and Planning, Jiangsu Normal University, Xuzhou 221116, China

**Keywords:** impervious surface area, self-training, deep forest, Sentinel-2, GaoFen-3, PolSAR

## Abstract

Impervious surface area (ISA) has been recognized as a significant indicator for evaluating levels of urbanization and the quality of urban ecological environments. ISA extraction methods based on supervised classification usually rely on a large number of manually labeled samples, the production of which is a time-consuming and labor-intensive task. Furthermore, in arid areas, man-made objects are easily confused with bare land due to similar spectral responses. To tackle these issues, a self-trained deep-forest (STDF)-based ISA extraction method is proposed which exploits the complementary information contained in multispectral and polarimetric synthetic aperture radar (PolSAR) images using limited numbers of samples. In detail, this method consists of three major steps. First, multi-features, including spectral, spatial and polarimetric features, are extracted from Sentinel-2 multispectral and Chinese GaoFen-3 (GF-3) PolSAR images; secondly, a deep forest (DF) model is trained in a self-training manner using a limited number of samples for ISA extraction; finally, ISAs (in this case, in three major cities located in Central Asia) are extracted and comparatively evaluated. The experimental results from the study areas of Bishkek, Tashkent and Nursultan demonstrate the effectiveness of the proposed method, with an overall accuracy (OA) above 95% and a Kappa coefficient above 0.90.

## 1. Introduction

With the rapid development of urbanization in the world, impervious surface area (ISA) has replaced the natural landscape as the main type of urban surface cover [1]. The substantial increase in ISA has caused a series of urban environmental problems, such as urban heat islands [2], urban flooding [3], water pollution [4] and so on. Therefore, timely and accurate estimation of urban ISA is of great significance for urban planning, sustainable development and urban environmental management.

Compared with less time-efficient manual measurements, multi-platform and multi-temporal remote sensing (RS) images facilitate the acquisition of ISA data. At present, the commonly used ISA extraction methods based on RS images can be roughly grouped into three categories: spectral mixture analysis (SMA), spectral index, and image classification and regression methods. The vegetation–impervious surface–soil (V–I–S) model proposed by Ridd to characterize the urban environment is the basis of the SMA method [5]. Wu and Murray [6] further divided impervious surfaces into low-albedo and high-albedo surfaces, then estimated the distribution of ISA using a linear SMA. These methods are good at dealing with mixed pixels, but, due to the variability of endmembers (within and between classes), the choice of endmembers plays a critical role in the accuracy of the results [7]. Although many scholars have proposed improved variants, the difficulty of endmember selection is still the main restriction on the large-scale application of this method [8,9,10]. On the contrary, spectral index-based methods have obvious advantages of general applicability, simplicity and efficiency. Therefore, a series of indices have been developed to extract ISA information using multispectral RS imageries, including the normalized difference impervious surface index (NDISI) [11], biophysical composition index (BCI) [12], perpendicular impervious surface index (PISI) [13], normalized difference built-up index (NDBI) [14], normalized built-up-area index (NBAI) [15] and modified NDISI [16]. However, the ISA extraction capabilities of these methods are always limited because of spectrum confusion phenomena (the same kind of objects with different spectral responses or different kinds of objects with the same spectral response). For instance, these methods always tend to confuse bare soil and ISA, particularly in arid and semi-arid areas. The image classification and regression methods that possess the advantage of separating ISA from other ground objects are mostly based on traditional supervised classification and regression methods, such as maximum likelihood [17,18], support vector machine (SVM) [19,20,21], random forest (RF) [22,23,24] and artificial neural networks (ANNs) [19,25,26]. More recently, researchers have also explored the performance of deep learning (DL) methods in ISA extraction. For example, convolutional neural networks (CNNs) have been introduced for ISA extraction, demonstrating promising ISA results from high-spatial-resolution, hyperspectral and multi-source RS images [27,28,29]. Although both shallow and deep supervised classification and regression methods are capable of reaching superior performances compared with the methods from the other two groups, manually collecting a large number of samples at good quality and representativeness is still always a challenging task in real scenarios, especially in large-scale coverage, multi-stage and long-term analysis cases.

Due to the limitations of optical images, many researchers have explored the use of multi-source RS data as a supplement, such as night-time light (NTL) [30,31], light detection and ranging (LiDAR) [18,28], synthetic aperture radar (SAR) and polarimetric SAR (PolSAR) data [22,32,33,34]. Among these, SAR has an advantage in ISA extraction by virtue of its all-time and all-weather backscattering characteristics, which are sensitive to geometric properties. The strong sensitivity of radar signals to man-made structures results in the different backscattering characteristics of ISA and other natural ground objects [20]. Therefore, SAR images were used to distinguish ISA from bare soil [22,34]. Guo et al. [33] compared the application of optical images and PolSAR data and demonstrated that the synergistic use of the two can improve urban ISA estimation accuracy. Zhang et al. [22] adopted a fusion method combining pixel-level and feature-level inputs to fuse optical and SAR imageries, which improved ISA estimation results. However, in data fusion, the classifier, the fusion level and the urban landscape all have a great influence on ISA extraction results [35,36]. In addition, PolSAR has richer polarization information than SAR, and studies have shown that more polarization can lead to better results [32,37]. Due to the special geographical attributes of arid and semi-arid areas, the spectral information of artificial ISA and surrounding objects is confusing. In previous research on ISA, the accuracy in arid and semi-arid areas was poor compared with that in humid and semi-humid areas [38,39].

In the machine-learning (ML) and pattern-recognition (PR) communities, active learning (AL) [40], semi-supervised learning (SSL) [41], instance transfer learning (ITL) [42] and few-shot learning (FSL) [43] are popular frameworks to face the challenges of learning from small samples. In contrast with AL, ITL and FSL, SSL exploits the large amount of unlabeled sample information in images to alleviate the dependence on manual annotation at no extra cost. In the field of urban RS, SSL is a common means of change detection [44,45,46], image classification [47,48] and building extraction [49,50]. In addition, SSL has also been used for long-term series ISA dynamics monitoring [51,52]. Self-training is an iterative SSL algorithm based on a specific classifier, which is simple to implement and requires no extra assumptions [53]. Generally, it trains the classifier with a small number of labeled samples and feeds the results into the next training session. In the past few years, self-training has been commonly used for the classification and segmentation of RS images [48,54,55,56,57]. Li et al. [55] proposed an image-classification and segmentation framework combining self-training and conditional random fields and proved that adding self-training improves classification accuracy with limited labeled training samples. Li et al. classified PolSAR images based on self-training and superpixels, which method showed good performance. Nevertheless, with this method, predictions are considered to be correct, which leads to its strong dependence on the performance of the classifier.

Deep forest (DF) is a non-neural-network (NN)-based DL model proposed by Zhou and Feng [58] based on an ensemble-learning approach. Compared with NN-based DL methods, a key advantage of the DF is its adaptive model complexity, which performs excellently even on small-scale data. In addition, it has far fewer hyper-parameters than DNN. Owing to these attractive advantages, the application of DF has received increasing attention in different fields in the past few years, such as biology [59], engineering [60] and RS [61]. At present, it has been successfully applied in object-detection [62], data-fusion [63,64] and image-classification tasks [61,65,66]. However, there is a lack of research on the application of DF in ISA extraction, especially with limited samples in arid areas.

Therefore, this paper proposes an ISA extraction method which uses a self-trained deep forest with limited labeled samples and multiple features of Sentinel-2 multispectral images and GaoFen-3 (GF-3) PolSAR images and applies it to obtain ISA extraction information for three major cities in arid Central Asia.

## 2. Materials and Methods

### 2.1. Study Area

Central Asia is the core of the Eurasian continent and a key node of the Belt and Road Initiative, located in one of the world’s largest arid and semi-arid inland areas. Different from humid and semi-humid areas, the vegetation coverage in Central Asia is low, and soil and rocks are widely distributed, which results in the poor applicability of optical image-based ISA extraction methods in this region. In addition, there are relatively few data on RS images and land cover types of Central Asian cities. Considering data availability and their properties, three cities, Nursultan, Bishkek and Tashkent, were selected as study areas (Figure 1). Nursultan is the capital city of Kazakhstan, located in the semi-desert steppe in north-central Kazakhstan. Bishkek is the capital and largest city of Kyrgyzstan and has a Mediterranean-influenced, humid continental climate. Located in northeastern Uzbekistan, Tashkent is the capital and largest city of Uzbekistan, as well as the most populous city in Central Asia.

### 2.2. Data

Chinese GF-3 is a C-band multi-polarization SAR imaging satellite with 12 imaging modes designed for various purposes, including not only traditional stripe and scanning imaging modes but also spotlight, hyperfine stripe and wave modes, etc. It has the characteristics of high resolution, large imaging width, multiple imaging modes and long-life operation. It can be used in marine contexts, target detection, disaster reduction, water conservancy and meteorology, etc. In the quad-polarization Stripmap I (QPSI) imaging mode of GF-3, the azimuth resolution is 8 m and the range resolution is 6–9 m. Sentinel-2A was designed by the European Space Agency (ESA) with an optical payload multi-spectral instrument (MSI) to sample 13 spectral bands at 10 m (bands 2, 3, 4, 8), 20 m (bands 5, 6, 7, 8a, 11, 12) and 60 m (bands 1, 9, 10) spatial resolutions. In this paper, the quad-polarized SAR images acquired by the QPSI imaging mode of GF-3 and imagery of Sentinel-2A in four bands of blue (B2), green (B3), red (B4) and NIR (B8) were employed to acquire image features. The quad-polarized SAR images were registered with optical images and resampled to 10 m. In addition, the Finer Resolution Observation and Monitoring of Global Land Cover with 10 m resolution (FROM-GLC10) product [67] was used as a reference and comparison for the extraction results. Table 1 shows the acquisition dates and spatial resolutions of the Sentinel-2A and GF-3 data that we used.

### 2.3. Method

#### 2.3.1. Feature Extraction

To extract ISA information more accurately, a number of candidate features considered in this study are shown in Table 2. Normalized difference vegetation indices (NDVIs) [68] and normalized difference water indices (NDWIs) [69] were obtained as spectral features from Sentinel-2 multispectral imagery. NDVI is an indicator used to assess vegetation cover, and NDWI is usually used for detecting water bodies. The two indices are expressed as below:(1)$$\mathrm{NDVI}=(b_{nir}-b_{red})/(b_{nir}+b_{red})$$
(2)$$\mathrm{NDWI}=(b_{green}-b_{nir})/(b_{green}+b_{nir})$$
where *b_red_*, *b_green_* and *b_nir_* stand for surface reflectance in the red, green and near-infrared bands, respectively.

As for spatial features, morphological attribute profiles with partial reconstruction (MAPPR) were employed. These can be used to model the attributes of different objects in remote sensing images and achieve good performance in classification tasks. The filtering in MAPPR is based on the merging of connected components of different gray levels [70,71]. An attribute *A* (such as area, diagonal length of a circumscribed rectangle, moment of inertia, etc.) is calculated for each connected component of the grayscale image for a given reference value *λ*. For the connected component *C_i_*, if the attribute satisfies the set condition (e.g., *A*(*C_i_*) > *γ*), the region remains unchanged; otherwise, it is merged into the adjacent regions with similar gray values. If the region is merged into a region with a lower (greater) gray level, the operation is one of thinning (thickening). In MAPPR, a pixel is reconstructed if the geodesic distance is smaller than the geodesic distance *d* in the mask *g* and *d* < ∞. MAPPR is obtained by applying a sequence of partially reconstructed attribute thinning and thickening to the gray image *f*:(3)$$\bar{AP}(f)=\left\{{\bar{\varphi }}_{n}\left(f\right),\ldots ,{\bar{\varphi }}_{1}\left(f\right),f,{\bar{\rho }}_{1}\left(f\right),\ldots ,{\bar{\rho }}_{n}\left(f\right)\right\}$$
where ${\bar{\varphi }}_{i}$ and ${\bar{\rho }}_{i}$ denote the proposed attribute thinning and thickening with criterion $\gamma_{i}$, respectively.

Three attributes have been considered in this paper, including area (the size of regions), standard deviation (the elongation of regions) and moment of inertia (related to the homogeneity of regions). These are used to show the unique structural characteristics of man-made objects, such as roads and block buildings, that differ from other surface covers. Furthermore, different attribute profiles can comprehensively characterize complex urban structures.

In fully polarimetric SAR data, the backscattering coefficient matrix can be described as ***S***:(4)$$S=\left(\begin{array}{cc} S_{HH} & S_{HV}\\ S_{VH} & S_{VV} \end{array}\right)$$
where *S_HH_*, *S_HV_*, *S_VH_* and *S_VV_* represent the backscattering coefficients of different polarizations.

Under the backscattering mechanism of monostatic radar, *S_HV_* = *S_VH_*. The adjoint matrix of a given matrix *A* will be denoted by *A**. Then, the coherence matrix ***T***_3_ can be described as:(5)$$T_{3}=\frac{1}{2}\left[\begin{array}{ccc} \langle {\left|S_{HH}+S_{VV}\right|}^{2}\rangle & \langle \left(S_{HH}+S_{VV}\right){\left(S_{HH}-S_{VV}\right)}^{*}\rangle & 2\langle \left(S_{HH}+S_{VV}\right)S_{HV}^{*}\rangle \\ \langle \left(S_{HH}-S_{VV}\right){\left(S_{HH}+S_{VV}\right)}^{*}\rangle & \langle {\left|S_{HH}-S_{VV}\right|}^{2}\rangle & 2\langle \left(S_{HH}-S_{VV}\right)S_{HV}^{*}\rangle \\ 2\langle S_{HV}{\left(S_{HH}+S_{VV}\right)}^{*}\rangle & 2\langle S_{HV}{\left(S_{HH}-S_{VV}\right)}^{*}\rangle & 4\langle {\left|S_{HV}\right|}^{2}\rangle \end{array}\right]$$

The *H*/*A*/*α* decomposition is based on an eigenvector analysis of the coherence matrix ***T***_3_, where *H* is the entropy, *A* is the anisotropy and $\bar{\alpha }$ is the polarimetric scattering parameter related to scattering direction.
(6)$$P_{i}=\lambda_{i}/\sum_{j=1}^{3}\lambda_{j}$$
(7)$$H=-\sum_{i=1}^{3}P_{i}\log_{3}P_{i}$$
(8)$$A=\frac{\lambda_{2}-\lambda_{3}}{\lambda_{2}+\lambda_{3}}$$
(9)$$\bar{\alpha }=\sum_{i=1}^{3}P_{i}\alpha_{i}$$
where $\lambda_{i}$ are the eigenvalues and $\alpha_{i}$ are the target scattering angles. *H*, *A* and $\bar{\alpha }$ are not affected by polarimetric azimuth offset and are rotationally invariant. These characteristic parameters are suitable for polarization classification and target recognition in urban and terrain relief areas.

The three-component Freeman polarimetric decomposition method was developed in 1998 [72]. It is based on a coherence matrix and decomposes a target into three scattering mechanisms. The scattering powers $P_{V}$, $P_{S}$ and $P_{D}$ of volume scattering, surface scattering and double-bounce scattering are, respectively, expressed as:(10)$$P_{V}=f_{V}$$
(11)$$P_{S}=f_{S}(1+{\left|\beta \right|}^{2})$$
(12)$$P_{D}=f_{D}(1+{\left|\alpha \right|}^{2})$$
where $f_{V}$, $f_{S}$ and $f_{D}$ correspond to the contributions of the volume, surface and double-bounce components, respectively; *β* and *α* are the surface and double-bounce scattering parameters, ranging from −1 to 1. This polarimetric decomposition method is based on the physical model of radar scattered echoes, which can be used to preliminarily determine the main scattering mechanism in backscattering. In this way, types of land cover can be effectively distinguished.

#### 2.3.2. Self-Trained Deep Forest

Generally, a DF consists of many base estimators to generate a cascade forest structure. Similar to an ANN, a cascade forest also shares a layer-by-layer structure. In its structure, each level of the cascade receives the feature information processed by the upper level and outputs its processing results at this level to the next level [73,74].

Consider the supervised learning problem of learning a mapping from the feature space ***χ*** to the label space **y**, where y={1,2,…,C}. Let $Z={\left[0,1\right]}^{C}$ and the training set $S=\{(x_{1},y_{1}),\ldots ,(x_{m},y_{m})\}$ be drawn independently and identically from the underlying distribution ***Γ***, where *m* is the size of the training set.

The deep forest model g:χ→y with T levels defined by a pair (h,f) can be formalized as:(13)$$g(x)=\underset{c\in \{1,\ldots ,C\}}{\arg\max}{\left[f_{T}(x)\right]}_{c}$$
where ${\left[f_{T}(x)\right]}_{c}$ is the *c*-th element of the label vector $f_{T}(x)$.

$f_{t}$ is the cascade of elements of forests $f=\{f_{i},\ldots ,f_{T}\}$ up to level *t*. At level t∈{1,…,T}, $f_{t}:\chi \to Ζ$ is defined as:(14)$$f_{t}(x)=\{\begin{array}{ll} h_{1}(x) & t=1\\ h_{t}([x,f_{t-1}(x)]) & t>1 \end{array}$$
where $h_{t}$ is the ensemble of forest $h=\{h_{1},\ldots ,h_{T}\}$ at level *t*. At every level *t*, $h_{t}(\cdot )$ and $p_{i}$ output a class vector $\left[p_{i}^{t},\ldots ,p_{C}^{t}\right]$, where $p_{i}$ is the prediction confidence of class *i* and the input of $h_{t}$ is $\left[x,f_{t-1}(x)\right]$, except that at level *t* = 1, its input is *x*.

After expanding a new layer, the entire cascaded forest will be estimated on the validation data. The training process will be terminated until there is no significant improvement in performance. Therefore, although the cascaded forest is able to adaptively determine the number of layers, it is prone to overfitting in the case of limited training samples.

In order to reduce overfitting and improve classification accuracy with a limited number of labeled samples, we propose a self-trained deep forest (STDF) method. A training set containing labeled and unlabeled samples will be inputted into the cascade forest, and the unlabeled instances will be labeled by iteration over the labeled samples. These pseudo-labeled samples can be used to train the model for higher performance in ISA extraction. As illustrated in Figure 2, the proposed STDF classification method is composed of three major steps:Input the labeled samples (initial training set) and unlabeled samples into the cascade forest with RF and ExtraTrees as the basic estimators in a certain proportion.Train the classifier and make predictions for unlabeled samples. The pseudo-labels with prediction probabilities greater than a threshold are added to the training set.Repeat the previous step. The classifier will continue iterating until the specified maximum number of iterations is reached.

## 3. Results and Discussion

### 3.1. Experimental Setup

Parameter Settings: (a) By verifying the accuracy of the number of 10–100 samples per class (10 groups in total) and the proportion of labeled samples to training samples (0.1–1.0), 30 training pixels per class, a total of 150, were randomly selected from labeled samples. Additionally, in the training set, the proportion of labeled samples was 0.8. (b) In verifying the extraction accuracy, the bright ISA and dark ISA were selected as impervious samples, while vegetation, water and bare land were selected as pervious samples. Then, 8000 test pixels per class, a total of 16,000, were randomly selected to examine the extraction accuracy for ISA. (c) The optimal parameters were determined by the grid search method, which optimizes the model by traversing the given parameters and using a cross-validation method. The number of estimators in each cascade layer is 2; the number of trees in each estimator is 125; and the maximum number of cascade layers is 25. In DF, the base classifiers are set as default by RF and ExtraTrees.Comparison with Other Methods: In this study, three classification methods, including RF, self-trained RF (STRF) and DF, were compared to evaluate the performance of the STDF algorithm. In order to test the superiority of the combination of the multispectral image features and PolSAR features proposed in this paper, the results for the use of multispectral image features alone were added for comparison. In addition, PISIs, for which the best performance was achieved with Sentinel-2 [75], NBAIs and land cover datasets (FROM-GLC10) were used for comparison. PISI and NBAI are, respectively, expressed as below [13,15]:
(15)$$\mathrm{PISI}=0.8192b_{blue}-0.5735b_{nir}+0.075$$
(16)$$\mathrm{NBAI}=\frac{b_{swir2}-b_{swir1}/b_{green}}{b_{swir2}+b_{swir1}/b_{green}}$$
where *b_blue_*, *b_nir_* and *b_green_* denote the surface reflectance in the blue, near-infrared and green bands, respectively, and *b_swir_*_1_ and *b_swir_*_2_ are two SWIR bands, satisfying *b_swir_*_2_ > *b_swir_*_1_.

Accuracy Assessment: To evaluate the performance for ISA extraction, ground reference data were collected by visual interpretation of high-spatial-resolution Google Earth images from the same period. The labeled samples were divided into five land-cover types (bright ISA, dark ISA, water, vegetation and bare land) and were evenly distributed in the study area. Subsequently, the training and testing samples were randomly selected among the labeled samples. The accuracy of ISA extraction was assessed by overall accuracy (OA), Kappa coefficient, commission error (CE) and omission error (OE) based on the confusion matrix. In addition, receiver operating characteristic (ROC) and area under ROC curve (AUC) were used for classification performance assessment.

### 3.2. Performance of PolSAR in ISA Enhancement

The polarimetric features applied in this paper were obtained by applying decomposition methods. These polarimetric features reflect the scattering mechanism of ground objects on the SAR bands. Figure 3 shows the ISA extraction results for NBAI based on multispectral bands and pseudo-color images of various polarimetric decomposition methods in the three study areas.

As illustrated in Figure 3, in the results for ISA extraction based on NBAIs, it can be seen that vegetation and ISA are well-differentiated, while bare land and ISA are easily confused due to the similar spectral information. Different from optical data, PolSAR reflects the scattering properties of ground objects and is sensitive to geometric properties. The PolSAR polarimetric features show that bare land and ISA have different backscattering characteristics which are easy to distinguish. The reason for this performance is the different scattering patterns for ISA and bare land [76]. However, the similar scattering characteristics of vegetation and ISA cause confusion between the two, including double-bounce scattering, volume scattering and surface scattering. Therefore, PolSAR backscatter information and multispectral information can complement each other.

Among these PolSAR decomposition methods, Freeman-3 and An & Yang-3 component decomposition are based on scattering models, and Cloude decomposition is based on target vector features. In general, the pseudo-color images of these three decomposition methods are consistent, reflecting three basic scattering mechanisms: surface scattering, double-bounce scattering and volume scattering. According to Figure 3, in the Freeman-3 decomposition results, ISA can be effectively distinguished. Compared with the other two methods, the results of the Freeman-3 decomposition method show more obvious differences between impervious surfaces and other types of ground objects and the urban structure is better reflected. This difference mainly depends on the surface roughness and geometry of ISA. Different from the other three decomposition methods, *H*/*A*/*α* decomposition explains the mechanism of SAR image data. In this decomposition, *α* is used to identify the scattering mechanism, the polarization entropy *H* reflects polarization information and the parameter *A* complements parameter *H*. When *H* increases to a higher value, *A* is used for the identification of the scattering mechanism [77]. With this method of decomposition, bare land and water are more clearly manifested. Therefore, Freeman-3 component decomposition and *H*/*A*/*α* decompositions were chosen as the polarimetric features obtained by the decomposition.

### 3.3. Results for ISA Extraction and Accuracy Assessment

Since the training samples were randomly selected and the selected samples were not necessarily uniformly distributed and representative, we performed 10 experiments with each number of labeled samples and the accuracy of results was evaluated and averaged. The results for different classification methods using different numbers of labeled training samples to extract ISA are shown in Figure 4. Accordingly, as the number of labeled training samples increased, the Kappa coefficients of four strategies improved for all three study areas, and the Kappa values for STDF were basically higher than those for the other three methods. As the number increased to 30, the Kappa values greatly improved. Additionally, learning curves for Kappa values for STDF always stayed at the highest level compared with DF, RF and STRF classifiers for all three study areas. In practice, more labeled training samples represent more time consumption. There is a trade-off between extraction accuracy and time consumption. Therefore, in the following experiments, we randomly selected 30 labeled samples per class.

The ISA extraction results are shown in Figure 5 and the accuracy assessments for the different methods are displayed in Table 3. From the accuracy assessments in Table 3, the OAs were all above 95% and the Kappa values were all above 0.90. In terms of visual inspection, our method performed well in all three study areas. In Figure 5, the confusion between ISA and barren land is obvious for the spectral-based method (PISI and STDF based on multispectral image features). In PISI and spectral-based STDF, some bare soil was wrongly classified as ISA. Conversely, more ISA was misclassified as pervious surface in PISI. Especially in Bishkek, misclassification was more serious. In addition, the OAs and Kappa values for the classification methods based on multispectral and PolSAR image features were all higher than the PISI and STDF results based on multispectral image features. This indicates that the combination of multiple features from multispectral and PolSAR images can be used as an effective means of ISA extraction.

In Bishkek and Nursultan cities, there were many cases where other ground objects were wrongly classified as ISA in the results generated by the other three methods based on multispectral and PolSAR data (STRF, DF and RF). However, in Tashkent city, ISA was more often classified as pervious surface. The above shows that STRF, DF and RF are unstable in ISA extraction given limited labeled samples, which may be due to the randomness of samples. Compared with the other three strategies, our method obtained higher ISA extraction accuracy values. Both OEs and CEs were at low levels for the three study areas. However, ISA omission for Tashkent was more severe out of the three study areas, which may have been due to more regions of badly mixed pixels. In Tashkent, the urban distribution is relatively scattered and the ISA is cross-distributed with other land types, such as bare soil and farmland. By visual inspection, the FROM-GLC10 ISA product showed poor accuracy for Bishkek due to a large number of ISA omissions. Though it showed better accuracy for Tashkent and Nursultan, there was still a fair amount of bare soil misclassified as ISA.

From Figure 5 and Figure 6, it can be seen that there were different degrees of confusion of ISA and bare land for these methods. With our method, ISA and bare land were better differentiated, while the overestimation of ISA was reduced. Especially in area B, STRF, DF, RF and spectral-based STDF methods obviously misclassified bare land as ISA. As illustrated in Figure 6, STRF, DF and RF had similar ISA extraction results. Compared with our method, STRF, DF and RF incorrectly classified part of the bare soil as ISA in area A and omitted ISA in areas C and D. The results for the A and C regions indicate that the proposed method has better ISA extraction performance in rural areas with mixed pixels. At the same time, from the results of areas C and D, it can also be seen that ISAs, such as roads and parking lots, can be effectively distinguished from bare land. Furthermore, with respect to the shadows of buildings in area D, the proposed method also had excellent results, especially compared with the spectral-based methods. This indicates the advantages of combining SAR with multispectral features in ISA extraction. With the FROM-GLC10 product, there was an underestimation of ISA, and omissions were manifested in relation to the construction area in area A and the roads in areas C and D. Therefore, our method not only has higher accuracy but is also more suitable for cities in arid regions than FROM-GLC10.

ROC analysis is a graphical approach for analyzing the performance of classifiers which uses true positive rates (TPRs) and false positive rates (FPRs). AUC is a single scalar value that complements ROC, where a greater area means better performance. The ROCs and AUCs of different methods for the three experimental areas are shown in Figure 7. From Figure 7, it can be seen that, compared with the other methods, the proposed STDF has a relatively high TPR and a low FPR and that it achieved the highest ROC value. This indicates that STDF has a better classification performance in cases of limited labeled samples.

The above has demonstrated the effectiveness of STDF. In order to make the experiment more complete, we additionally used Sentinel-2 multispectral images (Table 4) of the three experimental areas taken in three different periods in the same year. Then, we combined their features and the GF-3 PolSAR features to extract ISAs for the three areas using the proposed STDF method. The ISA extraction results for each experimental area are shown in Figure 8. Comparing with results shown in Figure 5, again, it is clear that our proposed STDF method for ISA extraction in arid urban areas is efficient with high accuracy using a limited number of samples.

## 4. Conclusions

In this paper, we have proposed an ISA extraction method using Sentinel-2 multispectral and GF-3 PolSAR images of limited labeled samples. Based on spectral, spatial and polarimetric features, the proposed STDF performed well in ISA extraction by learning from a small number of samples. The results for the multispectral and PolSAR feature- based method indicated better classification accuracy than the method based on multispectral image features, demonstrating the effectiveness of the proposed method. Compared with other classification methods, the STDF achieved a higher ISA extraction accuracy on the basis of fewer samples and showed a better performance for the three study areas of Bishkek, Nursultan and Tashkent. Furthermore, compared with PISI and the FROM-GLC10 product, our method reduced the confusion of ISA and bare land and improved the extraction accuracy of ISA. However, since the training samples for STDF were randomly selected, the randomness of the training samples had a certain influence on the extraction results, especially in urban suburbs with more mixed pixels. Since mixed pixels are unavoidable with limited spatial resolutions, higher-resolution remote sensing data should be used to obtain more accurate ISA results.

## Figures and Tables

**Figure 1 sensors-22-06844-f001:**
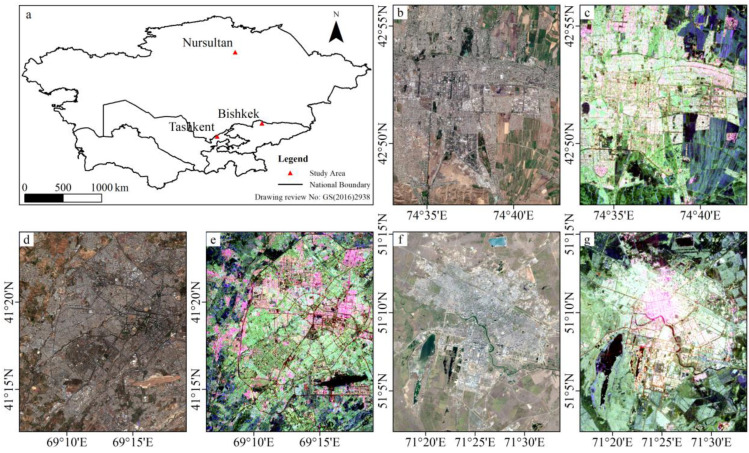
Locations and satellite images of the study areas. (**a**) Location of the study cites. (**b**) Sentinel-2 RGB image of Bishkek. (**c**) GF-3 Pauli-RGB image of Bishkek. (**d**) Sentinel-2 RGB image of Tashkent. (**e**) GF-3 Pauli-RGB image of Tashkent. (**f**) Sentinel-2 RGB image of Nursultan. (**g**) GF-3 Pauli-RGB image of Nursultan.

**Figure 2 sensors-22-06844-f002:**
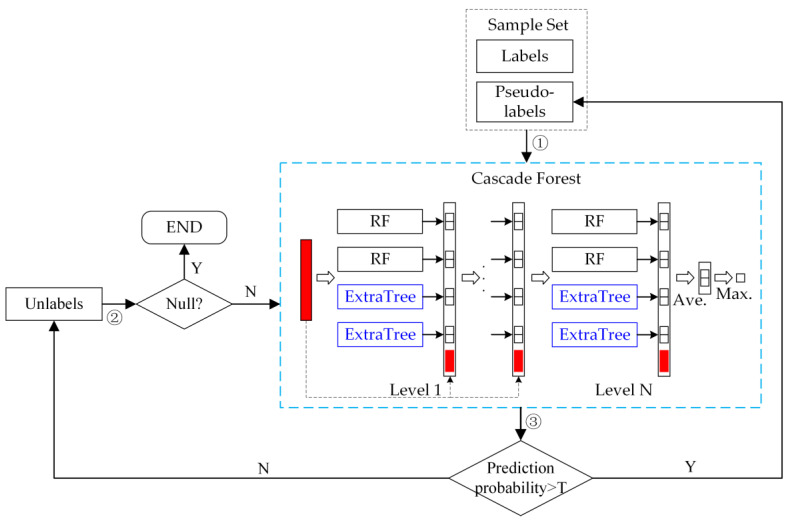
An overview of the proposed STDF.

**Figure 3 sensors-22-06844-f003:**
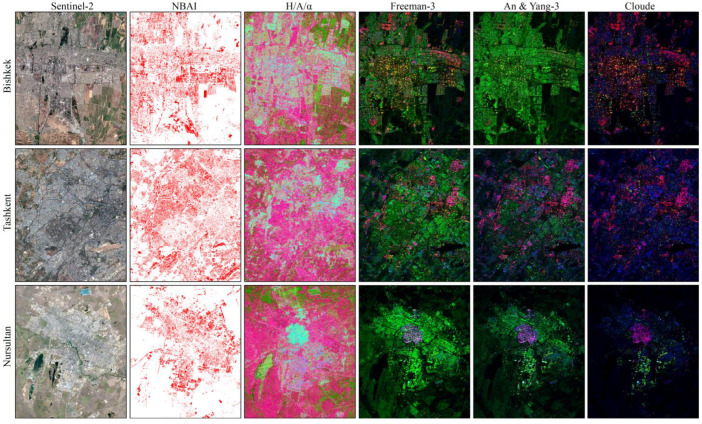
ISA extraction of NBAIs (red indicates ISA) and typical polarimetric decomposition comparison, RGB false color in H/A/α decomposition, Freeman-3 decomposition, An & Yang-3 decomposition and Cloude decomposition methods were, respectively, composited as: R (Entropy): G (Anisotropy): B (Alpha); R (Freeman_DBL): G (Freeman_VOL): B (Freeman_SURF); R (AnYang_DBL): G (AnYang_VOL): B (AnYang_SURF); and R (Cloude_T22): G (Cloude_T33): B (Cloude_T11).

**Figure 4 sensors-22-06844-f004:**
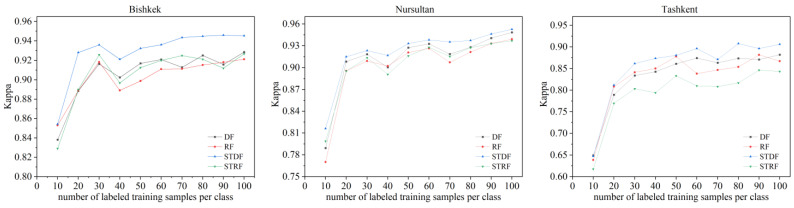
Kappa coefficients for four strategies using different numbers of training samples for the study areas.

**Figure 5 sensors-22-06844-f005:**
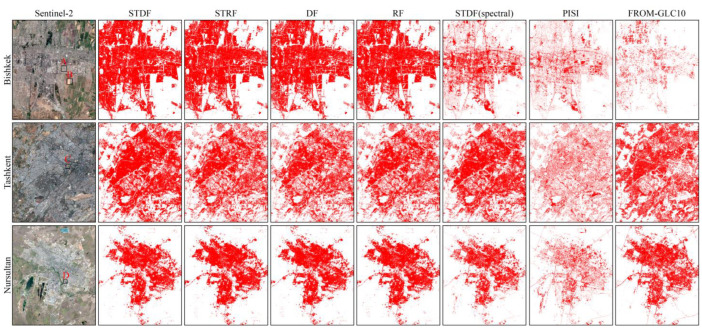
ISA extraction results for seven methods in three areas. Red indicates ISA. Areas A, B, C, and D are the areas with obvious differences between different methods.

**Figure 6 sensors-22-06844-f006:**
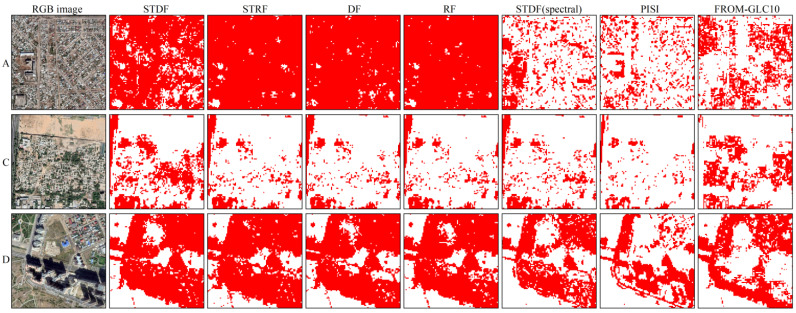
Comparison of ISA extraction results with a local zoom. The images in the first column were taken from Google Earth. Red indicates ISA. Areas A, C, and D correspond to those in Figure 5.

**Figure 7 sensors-22-06844-f007:**
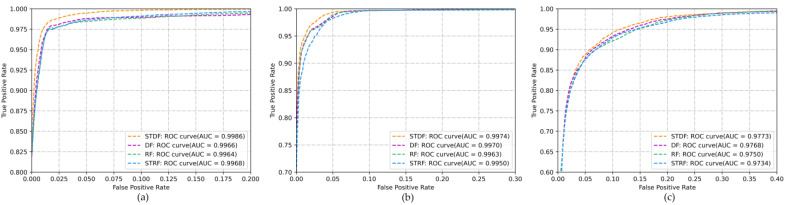
ROC curves and AUC values for different algorithms with respect to the dataset for the three regions: (**a**) Bishkek; (**b**) Tashkent; (**c**) Nursultan.

**Figure 8 sensors-22-06844-f008:**
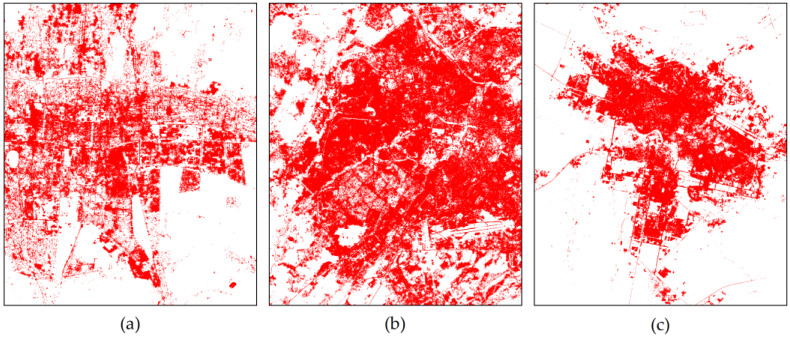
ISA extraction results based on Sentinel-2 data from different seasons and GF-3 data using the STDF method. Red indicates ISA. (**a**) Bishkek. (**b**) Tashkent. (**c**) Nursultan.

**Table 1 sensors-22-06844-t001:** Sentinel-2 and GF-3 data for the study areas used for ISA extraction.

Study Area	Sensor	Acquisition Dates	Pass	Resolution
Bishkek	Sentinel-2A	7 August 2017		10 m
	GF-3	9 November 2017	Ascending	8 m
Tashkent	Sentinel-2A	8 November 2017		10 m
	GF-3	25 October 2017	Descending	8 m
Nursultan	Sentinel-2B	17 September 2017		10 m
	GF-3	30 August 2017	Descending	8 m

**Table 2 sensors-22-06844-t002:** Feature sets and the number of features (*n*) per set.

Source	Feature Set	Descriptions	N
Sentinel-2A multispectral bands (B2, B3, B4 and B8)	Spectral features	NDVI	1
		NDWI	1
	Spatial features	Morphological attribute profiles with partial reconstruction	120
GF-3 quad-polarized SAR	Polarimetric features	T3 matrix	9
		H/A/α decomposition (entropy, anisotropy and alpha)	3
		Freeman decomposition (Freeman_DBL, Freeman_VOL and Freeman_SURF)	3

**Table 3 sensors-22-06844-t003:** Classification accuracy results for six methods.

		STDF	STRF	DF	RF	STDF (Spectral)	PISI
Bishkek	Kappa	0.9406	0.9249	0.9266	0.9174	0.7506	0.7394
	OA (%)	97.03	96.24	96.33	95.87	87.53	86.97
	OE (%)	2.16	2.39	2.61	2.58	9.59	17.56
	CE (%)	3.72	4.99	4.63	5.52	14.51	9.35
Tashkent	Kappa	0.9043	0.8110	0.8363	0.8474	0.8429	0.7819
	OA (%)	95.21	90.55	91.81	92.37	92.14	89.09
	OE (%)	6.38	11.58	12.71	12.88	10.13	16.65
	CE (%)	3.30	7.65	4.03	2.67	5.85	5.83
Nursultan	Kappa	0.9498	0.9266	0.9395	0.9280	0.9100	0.7595
	OA (%)	97.49	96.33	96.98	96.40	95.50	87.98
	OE (%)	0.98	0.61	0.88	1.34	3.63	11.74
	CE (%)	3.93	6.34	4.96	5.61	5.28	12.24

**Table 4 sensors-22-06844-t004:** Sentinel-2 data from three different periods in the study areas used for ISA extraction.

Study Area	Sensor	Acquisition Dates
Bishkek	Sentinel-2A	8 June 2017
	Sentinel-2A	7 August 2017
	Sentinel-2B	1 November 2017
Tashkent	Sentinel-2A	16 July 2017
	Sentinel-2A	2 September2017
	Sentinel-2A	8 November2017
Nursultan	Sentinel-2A	18 January 2017
	Sentinel-2A	18 May 2017
	Sentinel-2B	17 September 2017

## Data Availability

The FROM-GLC10 product used in this work is available at: http://data.ess.tsinghua.edu.cn/fromglc10_2017v01.html, accessed on 1 May 2022.

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
