# Peer review of "Self-Trained Deep Forest with Limited Samples for Urban Impervious Surface Area Extraction in Arid Area Using Multispectral and PolSAR Imageries"

_sensors, 2022, doi:10.3390/s22186844_

Round 1

Reviewer 1 Report

   Impervious surface area (ISA) has been recognized as a significant indicator for evaluating the level of urbanization and the quality of the urban ecological environment. ISA extraction methods based on supervised classification usually rely on a large number of manually labeled samples, which is practically a time-consuming and labor-intensive task. Furthermore, the extraction accuracy of ISA using optical images is always limited by the urban landscape complexity, heterogeneity and spectrum confusion facts, especially in the arid areas whereas the man-made objects are easily confused with bare lands. To tackle the above issues, a self-trained deep forest (STDF) based ISA extraction method is proposed by exploiting the complementary information between multispectral and polarimetric synthetic aperture radar (PolSAR) images using limited samples. In detail, this method consists of three major steps. First, multi-features, including spectral features, spatial features and polarimetric features are extracted from Sentinel-2 multispectral and Chinese GaoFen-3 (GF-3) PolSAR images; Secondly, deep forest (DF) model is trained in a self-training manner using a limited number of samples for ISA extraction; Finally, ISA for three major cities located in Central Asia are extracted and comparatively evaluated. The experimental results from the study areas of Bishkek, Tashkent and Nursultan demonstrate the effectiveness of the proposed method, with over- all accuracy (OA) above 95% and Kappa coefficient above 0.90.

 The paper is interesting and is expected to achieve high practical values. However, some issues need to be properly addressed, the major concerns are:

(1) The abstract is too tedious, the authors give too much statement on the problems that the existing ISA extraction methods exist. The authors should simply that.

(2) The methodolgy section does not well present the innovative work,the author simply use the existing methods on feature extraction and Deep Forest based ISA extraction, the methods are not the authors proposed. So what's your innovative work? the authors should specify it.

(3) The experiment is not enough. The authors simply use OA, Kappa, OE and CE for objective evaluation of the proposed method with others. However, these parameters are achieved on the single result of each image. PR curves should be acquired through Monte-Carlo simulation to better validate the superiority and effectiveness of your method, detailed simulation can be referred to: [1]https://ieeexplore.ieee.org/document/9631208,

[2] https://ieeexplore.ieee.org/document/9528903,

[3] https://ieeexplore.ieee.org/document/8812890.

Reviewer 2 Report

Dear Authors , please re-shape your experiment, re-think the methodology a bit and eliminate MAPPR factor. Sentinel-2 plus GF-3 will be enough for clear and successful publication. Please look at the attached file.

Round 2

Reviewer 1 Report

Some questions have not been carefullu addressed, it must be improved.  The authors did not read the papers, but claim they are not valiad for SAR classification, this is not correct. ROC curves and AUC values should be added to the experiment to better valiadate the superiority and effectiveness. Detailed simulation can be referred to:

[1]https://ieeexplore.ieee.org/document/9631208,

[2] https://ieeexplore.ieee.org/document/9528903,

[3] https://ieeexplore.ieee.org/document/8812890.OA, Kappa, OE and CE for objective evaluation of the proposed method with others. However, these parameters are achieved on the single result of each image. PR curves should be acquired through Monte-Carlo simulation to better validate the superiority and effectiveness of your method, detailed simulation can be referred to:[1]https://ieeexplore.ieee.org/document/9631208,[2] https://ieeexplore.ieee.org/document/9528903,[3] https://ieeexplore.ieee.org/document/8812890.experiment is not enough. The authors simply use OA, Kappa, OE and CE for objective evaluation of the proposed method with others. However, these parameters are achieved on the single result of each image. PR curves should be acquired through Monte-Carlo simulation to better validate the superiority and effectiveness of your method, detailed simulation can be referred to:[1]https://ieeexplore.ieee.org/document/9631208,[2] https://ieeexplore.ieee.org/document/9528903,[3] https://ieeexplore.ieee.org/document/8812890.Response 3: We thank the reviewer very much for the comment. Based on reviewer’s comment, we have consulted the relevant literature on methods for evaluating learners such as PR curves and ROC curves. PR curves are typically used in binary classification to study the output of a classifier. In this paper, the land cover was divided into five categorie

Reviewer 2 Report

Point 4: Only ONE Sentinel-2 image per site used? You should/have to increase the number of optical images! At least 2-3 taken in different seasons.

Response 4: We thank the reviewer very much for the constrctive comment. In this study, the spectral features and spatial features of optical images were used. And spectral features were used primarily to distinguish impervious surface area (ISA) from vegetation. We can easily obtain Sentinel-2 images of multiple periods, however, due to the limitation of data acquisition of GF-3 images, we cannot obtain the corresponding GF-3 images of the same period.

* Second round reviewer's comment on that question:  Please pay attention that impervious surfaces in big cities are not changing very quickly. We can assume that the same acreage can be attributed for the city for a given year (or longer time perid). It depends on the development rate ...  The scientists should seek the solutions which are: fast, simple, cheap and reliable... Please take into account 3-4 Sentinel-2 images registered in different seasons. It is not necessary to have the same quantity of GF-3 images. You look for the alternatives, isn't it ? 

Point 15: “The PolSAR polarimetric features show that bare lands and ISA have different backscattering characteristics, which are easy to distinguish.” Please explain in this place "WHY" ?  * I'm not convinced by the explanation.  As well as in case of point 17.
